# Advances in Neuro-Oncological Imaging: An Update on Diagnostic Approach to Brain Tumors

**DOI:** 10.3390/cancers16030576

**Published:** 2024-01-30

**Authors:** Paniz Sabeghi, Paniz Zarand, Sina Zargham, Batis Golestany, Arya Shariat, Myles Chang, Evan Yang, Priya Rajagopalan, Daniel Chang Phung, Ali Gholamrezanezhad

**Affiliations:** 1Department of Radiology, Keck School of Medicine, University of Southern California, 1500 San Pablo St., Los Angeles, CA 90033, USA; panizsabeghi1992@gmail.com (P.S.); eyang4@dhs.lacounty.gov (E.Y.); priya.rajagopalan@med.usc.edu (P.R.); daniel.phung@med.usc.edu (D.C.P.); 2School of Medicine, Shahid Beheshti University of Medical Sciences, Tehran 1985717411, Iran; panizzarand@gmail.com; 3Department of Basic Science, California Northstate University College of Medicine, 9700 West Taron Drive, Elk Grove, CA 95757, USA; sina.zargham7387@cnsu.edu; 4Division of Biomedical Sciences, Riverside School of Medicine, University of California, 900 University Ave., Riverside, CA 92521, USA; batis.golestany@medsch.ucr.edu; 5Kaiser Permanente Los Angeles Medical Center, 4867 W Sunset Blvd, Los Angeles, CA 90027, USA; arya.x.shariat@kp.org; 6Keck School of Medicine, University of Southern California, 1975 Zonal Avenue, Los Angeles, CA 90089, USA; mylescha@usc.edu

**Keywords:** central nervous system (CNS), magnetic resonance elastography (MRE), magnetic resonance fingerprinting (MRF), MR perfusion imaging, neuro-oncology imaging, positron emission tomography (PET)

## Abstract

**Simple Summary:**

In the realm of neurology, advanced imaging tools play a crucial role as critical endpoints in clinical trials. While magnetic resonance imaging (MRI) serves as a primary diagnostic tool, it exhibits limitations in specific scenarios. Ongoing research in neuro-oncological imaging aims to address these limitations. Our review explores the latest advancements in imaging modalities for neuro-oncology, highlighting the accuracy and competence of each modality. These include PET tracers and radiolabeled amino acids, PET/MRI, radiomics, deep learning, MR perfusion imaging, MR fingerprinting, MR spectroscopy imaging, MR elastography, and intra-operative ultrasound techniques. The focus is on the potency of these modalities in diagnosis, cancer staging, prognosis, and post-treatment evaluation, ultimately enhancing accuracy and effectiveness in managing brain tumors.

**Abstract:**

This study delineates the pivotal role of imaging within the field of neurology, emphasizing its significance in the diagnosis, prognostication, and evaluation of treatment responses for central nervous system (CNS) tumors. A comprehensive understanding of both the capabilities and limitations inherent in emerging imaging technologies is imperative for delivering a heightened level of personalized care to individuals with neuro-oncological conditions. Ongoing research in neuro-oncological imaging endeavors to rectify some limitations of radiological modalities, aiming to augment accuracy and efficacy in the management of brain tumors. This review is dedicated to the comparison and critical examination of the latest advancements in diverse imaging modalities employed in neuro-oncology. The objective is to investigate their respective impacts on diagnosis, cancer staging, prognosis, and post-treatment monitoring. By providing a comprehensive analysis of these modalities, this review aims to contribute to the collective knowledge in the field, fostering an informed approach to neuro-oncological care. In conclusion, the outlook for neuro-oncological imaging appears promising, and sustained exploration in this domain is anticipated to yield further breakthroughs, ultimately enhancing outcomes for individuals grappling with CNS tumors.

## 1. Introduction

In the field of neurology, imaging plays a central role in diagnosis, predicting prognosis, and assessing treatment response for central nervous system (CNS) tumors. Evaluation through imaging may also serve as a crucial substitute for endpoints in clinical trials. The continuous evaluation and discovery of new therapeutic agents, including immunotherapy, underscores the central objective of neuro-oncologic imaging, which is the accurate evaluation of disease progression and the identification of treatment-related changes [1]. 

Malignant brain tumors can be categorized into two broad groups: metastatic tumors, which arise from locations outside the brain, and primary tumors, which originate within the brain tissue itself and its surrounding meninges. Metastatic brain tumors most commonly originate from the lungs, breasts, and skin, particularly melanoma [2]. Over 100 distinct primary CNS tumor cell types contribute to different histopathologies, with each demonstrating a unique set of clinical presentations, treatment options, and potential outcomes. In addition to histology and immunohistochemistry, substantial advancement in molecular diagnostics has allowed for histogenetic classification of various types and subtypes of these tumors, as described in the recent fifth edition of the WHO classification of brain tumors. In a study spanning 2016–2020, the average age-adjusted incidence of all malignant and non-malignant CNS tumors was 24.83 per 100,000 people. In that study, roughly 27.9% of all CNS tumors were found to be malignant and 72.1% were categorized as non-malignant or benign. Gliomas constituted 26.3% of all tumors. Among the primary malignant tumor histopathologies, glioblastoma (GBM) was the most frequently occurring, constituting 14.2% of all tumors and 50.9% of all malignant tumors. Conversely, meningioma (Figure 1) was the most common non-malignant tumor, accounting for 40.8% of all tumors and 56.2% of all non-malignant tumors [3].

The prognosis for patients with brain tumors, especially high-grade neoplasms, remains poor despite conventional treatments like surgery, radiotherapy, and chemotherapy. The complex and diverse nature of these tumors, along with frequent recurrence near the primary site, complicates their management [2].

To better facilitate accurate diagnosis and effective treatment planning, it is valuable to differentiate malignant and benign CNS tumors. Magnetic resonance imaging (MRI) serves as the main imaging modality for diagnosis and follow-up monitoring in patients with CNS tumors. However, conventional structural MRI remains limited in certain capacities and situations, including an inability to discern the full extent of infiltrative tumors (such as gliomas) and difficulty discriminating between neoplastic and non-neoplastic processes, particularly in the post-treatment setting (such as radiation necrosis after radiotherapy) [4]. Accordingly, neuro-oncologic imaging research has focused on addressing these shortcomings.

Here, our objective is to review the latest advancements in various imaging modalities utilized in neuro-oncology and to delve into their influences on diagnosis, cancer staging, prognosis, and post-treatment evaluation.

### 1.1. PET Tracers and Radiolabeled Amino Acids

Although structural imaging with MRI and computed tomography (CT) provide excellent image resolution and anatomical localization of brain tumors, supplemental molecular imaging using positron emission tomography (PET) imaging with radiotracers can provide vital details about the metabolic and proliferative activity of various cancers. Significant advancements have been made in the field of radiotracers and their utilization in clinical settings.

PET radiotracers have become an increasingly popular form of imaging due to their extensive capacity in identifying never-before-seen tumor activity in PET. One of the most famous and widely used radiotracers is 18-F-fluorodeoxyglucose (18F-FDG), a glucose analog. This radiotracer is widely used due to its proven efficacy in crossing the blood–brain barrier (BBB) with ease and its ability to tag highly metabolic areas, including tumors [5]. Although FDG is incredibly beneficial for tumor identification throughout the body, it remains particularly limited in the brain, especially due to the high level of glucose uptake in normal brain tissue, making it difficult to distinguish between normal and pathologic tissue [6]. 

Furthermore, PET radiotracer limitations become more pronounced when imaging patients throughout various treatment stages. Since treatment for brain cancer may impact tissue surrounding the tumor itself, radiotracers can sometimes tag these areas, rendering it difficult for radiologists to distinguish between the progression of cancer vs. treatment-related changes in brain tissue. To address this, various other radiotracers utilize tagged amino acids, rather than glucose, to achieve a more specific uptake pattern on the PET scan. One common amino acid radiotracer is [18F]-fluoroethyltyrosine ([18F]FET), which demonstrates decreased uptake by normal brain tissue when compared to 18F-FDG, thereby providing a greater distinction between normal and cancerous brain tissue [7]. 

Research regarding new and advanced radiotracers has emerged, further proving the substantial utility of this technology. Recently, new protein markers have demonstrated an increased specificity for brain cancer, as well as an increased ability to cross the BBB. One such protein is [18F] PARPi, which is a protein overexpressed in cancer cell nuclei. A substantial advantage of this radiotracer when compared to FDG is that its uptake is completely independent of metabolism. This decreases the likelihood of its uptake by other healthy, highly metabolic tissue in the brain [6]. Another tracer, known as fibroblast activation protein inhibitor (FAPI), tags an inhibitor known to be upregulated in some cancers. Early research studies have shown that although the inhibitor is not upregulated in diffuse astrocytomas, it is upregulated and traceable in isocitrate dehydrogenase (IDH)-wildtype GBM (Figure 2) and high-grade IDH mutant astrocytomas (Figure 3 and Figure 4) [8]. 

As new treatments emerge for cancer patients, new imaging tools must be used to better differentiate between cancerous tissue and recovering tissue. One such field is stereotactic radiosurgery, wherein surgeons irradiate specific brain tissues in a targeted manner, avoiding injury to surrounding healthy tissue [9]. Various studies assessing post-treatment tumor recurrence have been conducted, and one emerging radiotracer that has proven successful is an amino acid radiotracer known as [11C] Methionine, which is discussed further in the following section [10].

### 1.2. PET and PET/MR in Neuro-Oncology

PET and MRI can serve as complementary imaging modalities, each with their own strengths. Conventional MRI is known for its ability to provide high-resolution structural images of the brain, offering exceptional tissue contrast [11]. As such, it is an invaluable imaging modality for many non-traumatic anatomical neurological conditions, including epilepsy and tumors [12]. A particularly valuable aspect of MRI is diffusion-weighted imaging, which can be utilized to evaluate cell density, estimate tumor grade and extent, guide surgical resection and radiotherapy treatments, and assist in forecasting mortality outcomes [13].

On the other hand, PET focuses on delivering physiological data, offering insights into brain metabolism and functional processes. In oncological applications, PET serves multiple roles, from initially differentiating high-grade from low-grade tumors to guiding biopsy site selection and the extent of resection and radiation therapy at diagnosis. Post treatment, it aids in assessing either recurrence or the potential transformation to higher-grade malignancy [14,15,16]. PET imaging can employ different tracers, including FDG or amino acid tracers, each with distinct advantages. FDG, a glucose analog, allows for the detection of differences in glucose metabolism between malignant and physiological cells [13]. However, FDG-PET’s ability to assess tumor margins can be limited due to high uptake in normal brain parenchyma. In contrast, amino acid PET provides better visualization of tumor borders because normal brain tissue does not exhibit increased amino acid uptake [14,17]. Nonetheless, the combined use of PET and MRI can mitigate the limitations inherent to each individual modality.

When employed in tandem, PET/MRI offers a number of compelling advantages, including enhanced soft tissue contrast and a reduction in ionizing radiation exposure [11,12]. Head movement during PET scanning can disrupt proper attenuation correction or result in incorrect alignment of PET information with MR images. To address this, motion tracking based on MR imaging can be employed to reposition the PET data accurately [11,18]. The decrease in radiation is particularly beneficial for the pediatric population, where CNS cancer is a leading cause of death. Utilizing PET/MRI significantly reduces the cumulative radiation dose for these vulnerable patients [14]. Overall, the combination of PET and MRI technologies not only facilitates an effective initial characterization of disease but also allows for meticulous monitoring of disease progression and the evaluation of treatment effectiveness. Together, PET and MRI provide a comprehensive, multidimensional view of the brain, encompassing both its structural intricacies and dynamic activities [18]. 

PET/MRI can provide vital information in the challenging landscape of neuro-oncology, such as in the diagnosis and management of gliomas. Gliomas represent approximately 80% of malignant brain tumors and are notorious for their high rates of recurrence and poor survival outcomes [19,20]. Hence, distinguishing between recurrence and post-treatment changes is critical. Conventional MRI often faces challenges in this distinction due to the similar appearance of tumor recurrence and radiation necrosis [19,21]. PET/MRI, particularly with the use of C11-methionine as a tracer, outperforms both MRI and CT alone in this regard [19]. Studies have found that the diagnostic accuracy, sensitivity, and specificity of hybrid C11-MET-PET/MRI are superior to those of MRI alone [22,23,24]. In fact, combined PET/MRI achieved an impressive diagnostic accuracy rate of 95%, compared to 63% for PET and 82% for MRI [25]. Additionally, Deuschl et al. [22] reported a sensitivity of 97.14% and a specificity of 93.33% for 11C-MET-PET/MRI, further supported by Pauleit et al. [26] with a reported sensitivity of 93% and specificity of 94% for dual MRI/FET PET. Adding further weight to these findings, the integration of PET/MRI with parallel MRI (pMRI) delivered a remarkable 100% diagnostic sensitivity and specificity in differentiating between tumor progression and radiation necrosis post-treatment [21].

While gliomas are the predominant concern in malignant brain tumors, primary CNS lymphomas (PCNSLs) present a different set of challenges. PCNSLs constitute 1–5% of all brain tumors and are more commonly observed in immunocompromised patients [27]. Early diagnosis is crucial for initiating chemotherapy, highlighting the vital role of imaging in the management of these patients. In a study of patients over 60 years old with PCNSLs, baseline cerebellar metabolism and metabolic tumor volume (sumMTV) assessed via [18F] FDG PET/MRI were significant predictors of chemotherapy response. Additionally, larger tumor volumes at diagnosis were associated with poorer overall survival and early death [28]. 

Emphasizing the crucial role of PET/MRI in the post-treatment management of PCNSLs, [18F] FDG PET/MRI proves valuable in distinguishing between gliomas and PCNSLs, thereby aiding in the selection of appropriate treatment. Despite differences in their MRI appearances, there can be significant overlap in imaging appearance which can make diagnosis challenging. A multiparametric approach that utilizes 18F-FDG PET/MRI has the potential to differentiate high-grade gliomas (HGGs) from PCNSLs [29].

Dual PET/MRI’s expanding applications encompass other types of CNS tumors, including the diagnosis and treatment of meningiomas. MRI is the current diagnostic gold standard for meningioma, although it has limitations, particularly in post-surgical and post-radiotherapy settings [30]. Recent advancements suggest that MRI combined with [86Ga]-DOTATATE PET can enhance meningioma diagnosis, treatment planning, and post-treatment evaluation. The combined approach demonstrates superior differentiation of meningioma from healthy tissue and post-surgical changes [30,31]. [86Ga]-DOTATATE PET achieved a sensitivity of 97.6%, with a standardized uptake value (SUV) threshold of 2.3, and a specificity of 86.1%, with an SUV ratio referencing the pituitary gland (SUVRpit) threshold of 0.3 [30]. This suggests that the technique could be an important tool for enhancing diagnostic accuracy in the management of meningiomas. MRI is often less effective in detecting smaller meningiomas, with a sensitivity of 74% for lesions < 0.5 cm^3^ [30,32], a gap that [86Ga]-DOTATATE PET/MRI can help to fill [32]. 

Lastly, the brain is a common site for metastasis of extracranial tumors (Figure 5 and Figure 6) and many lesions are treated with radiosurgery [33]. However, distinguishing between recurrent brain metastases and radiation necrosis is again a challenge for conventional MRI. This issue can be addressed through the use of PET/MRI combined with radiomics. A study by Lohmann et al. demonstrated that the diagnostic accuracy for discerning recurrent brain metastases from radiation necrosis could be elevated to nearly 90% by integrating textural features from both CE-MRI and static FET PET scans [34]. Figure 7 shows an example for recurrent brain metastases in a renal cell cancer patient post radiation. Also, Johannessen et al. [33] demonstrated 18F-FACBC could be a valuable tool for the early detection of easily overlooked brain metastases.

Ultimately, PET/MRI presents multiple benefits across various phases of neuro-oncological conditions. Brendle et al. [35] reported an 85% diagnostic accuracy for brain tumors, along with a sensitivity of 78% and a specificity of 89%. The study also highlighted its value in tracking disease progression, noting a nearly 100% positive predictive value, 93% sensitivity, and 95% specificity. Based on current research, PET/MRI has the potential to substantially impact patient care by clarifying unclear treatment outcomes. In their cohort, clinical management was re-evaluated in 53% of cases upon detecting signs of disease progression [35]. 

Nonetheless, the use of PET/MRI comes with its own set of challenges, such as high costs and restricted availability, in addition to the likelihood of false positive results in cases with inflammation, infection, or post-surgical changes [20]. Acquiring and interpreting PET/MR imaging studies also implicitly necessitates either two separate teams for PET and MR imaging or a single specialized team with training in both modalities [18].

### 1.3. Radiomics and Deep Learning

Radiomics involves the extraction of subvisual, quantitative data from routine medical images, such as MRI or PET, to form a 3D tumor phenotype. The radiomics workflow includes data acquisition, image pre-processing, tumor segmentation, feature extraction and selection, and model generation [36]. Closely related to this concept is radiogenomics, which correlates genetic mutation status with radiologic features. Deep learning methods such as convolutional neural networks (CNN) are a form of machine learning that imitate human cognition and are often used in the radiomics pipeline for feature selection and modeling using various classifiers [37]. 

In neuro-oncology, radiomics and deep learning have shown potential to aid in diagnosis, prognostication, treatment response monitoring, and determining tumor biomarkers and genomics. Although numerous radiomics and deep learning studies have shown promising results, these methods are not commonly used in clinical trials and have yet to be used in clinical practice. Obstacles to clinical adoption include the lack of biologic correlation of radiomics features as well as lack of generalizability and reproducibility between different sites and scanners. The future of radiomics and machine learning in neuro-oncology is dependent on overcoming these issues [38].

Thus far, radiomics features have successfully been used to differentiate GBM from solitary metastasis [39,40] and GBM from PCNSL [41]. As most radiomics studies focused on two-class classification, Priya et al. [42] demonstrated a three-class classification radiomics model that can differentiate GBM, metastasis, and PCNSL is also possible. In a recent study, Bathla et al. [43] compared the performance of machine learning and deep learning pipelines in three-class classification with the highest-performing deep learning pipeline having an area under the curve (AUC) of 0.854 on external validation. Three-class classification is more likely to have clinical utility and generalizability [42].

More recently, Stadbauer et al. [44] developed a radiomics and deep CNN model that differentiated GBM and brain metastasis based on oxygen metabolism data extracted from MRI. Using the parameters of cerebral metabolic rate of oxygen (CMRO2) and tissue oxygen saturation (mitoPO2), these diagnoses were differentiated more accurately than those made by radiologists. Malik et al. [45] found radiomics features that accurately differentiated low-grade gliomas (LGGs) from the peritumoral region (PTR) of GBM, which are often difficult to distinguish with visual inspection. Differentiating LGGs from GBM PTR could potentially aid in reducing the tumor volume that undergoes radiation treatment [45]. 

Another utility of radiomics is in determining the primary source of different types of metastases [46]. Ortiz-Ramon et al. [47] demonstrated that a radiomics model using 3D texture features differentiated lung cancer metastasis from breast cancer metastasis (AUC = 0.963) and lung cancer metastasis from melanoma metastasis (AUC = 0.936) with high accuracy. Differentiating primary tumors from brain metastases can help prevent delays in diagnosis and treatment.

Deep learning has also been demonstrated to be beneficial for real-time intra-operative diagnosis. Shen et al. [48] used near-infrared fluorescence imaging combined with a deep CNN (FL-CNN) to diagnose gliomas during surgery and compared the results to histologic examination, the current standard of practice. At high levels of specificity (>80%), the FL-CNN had higher sensitivity and also corrected over 70% of the neurosurgeons’ errors. The study demonstrates the potential for deep learning models to improve neurosurgery outcomes by enhancing intra-operative diagnosis [48].

Furthermore, radiomics, deep learning, and radiogenomics have shown great potential in improving survival prediction, grading, and determining the genetic status of gliomas [49,50,51,52]. Although stereotactic brain biopsy is the current gold standard for diagnosis and classification, it does not always capture the heterogeneous nature of gliomas. Therefore, radiomics and radiogenomics have the potential to non-invasively determine genetic status and prognosis via a more complete, “virtual biopsy”, which would also aid in more selective chemotherapy and immunotherapy.

Multiple studies have used MRI-derived radiomics features to accurately predict overall survival (OS) [49,53]. Kickingereder et al. [53] designed and created an artificial neural network (ANN) that better predicted overall survival than the criteria for assessing response in neuro-oncology, known as the Response Assessment in Neuro-Oncology (RANO).

Radiogenomics also provides prognostic value. Radiomics features with deep learning models have accurately predicted the genetic status of low-grade and high-grade gliomas, including IDH status, PTEN status, 1p19q-codeletion status, and the status of MGMT promoter methylation [54,55,56,57,58,59]. Choi et al. [60] developed a potentially generalizable combined deep learning and radiomics model that accurately predicted IDH mutation status in gliomas with multiple datasets. Yogananda et al. [61] used a deep learning model for T2WI MRI that predicted MGMT promoter methylation status with a 94.73% mean cross-validation accuracy. Wang et al. [62] used DCE-MRI and DWI radiomics features to forecast IDH mutation status and VEGF expression in gliomas, achieving an AUC of 0.909, 0.880, and 0.842 in external validation groups. 

Recently, Liu et al. [63] used a radiogenomics model that used radiomics features to predict immune cell infiltration (ICI), a tumor microenvironment biomarker, in GBM, and provide additional prognostication value. Eleven radiomics features were used to differentiate tumors with varying ICI scores which aided in prognostication. Lastly, van der Voort et al. [64] developed a CNN that simultaneously predicted IDH mutation status (AUC 0.90), 1p/19q co-deletion status (AUC 0.85), tumor grade (AUC 0.81), and tumor segmentation (Dice score 0.84) for gliomas. This represents a unique deep learning method that can answer multiple important clinical questions at once. 

Radiomics features and machine learning methods have been demonstrated to accurately differentiate between tumor progression and treatment-related changes or pseudoprogression, which has long been a challenge for radiologists [65,66,67]. Kim et al. [68] developed a multiparametric radiomics model (AUC 0.90) that included data from T1WI post-contrast, FLAIR, ADC, and cerebral blood volume. This model performed significantly better than radiomics models that only used conventional MRI (AUC 0.76) or ADC alone (AUC 0.78) and also performed superiorly in external validation (AUC 0.85). 

Recently, Müller et al. [69] used radiomics features in conjunction with FET PET parameters, specifically TBRmean (Tumor-to-Background Ratio, Mean) and TBRmax (Tumor-to-Background Ratio, Maximum), to differentiate between tumor progression and treatment-related changes with high accuracy (AUC 0.92). Prasanna et al. [70] also used COLLAGE features, a type of radiomics feature, to differentiate between radiation necrosis and tumor recurrence in primary and metastatic brain tumors using T1WI contrast-enhanced imaging. Zhang et al. [71] used a radiomics model based on multiparametric MRI, which included DWI and arterial spin labeling, which effectively distinguished between the recurrence of glioma and radiation necrosis with an AUC of 0.96 and performed better than the conventional MRI model (AUC 0.88). 

Radiomics models have also shown utility in predicting the response to various treatments, including immunotherapies and anti-angiogenic therapies. Li et al. [72] recently developed a radiomics model that evaluated the response to a combination therapy consisting of anlotinib, an anti-angiogenic drug, and temozolomide for recurrent gliomas. Being able to differentiate between patients with a good response to treatment and those with a poor response can help prevent delays in targeted treatments. George et al. [73] sought to create a radiomics model with the aim of predicting both progression-free survival (PFS) and overall survival in glioma patients treated with durvalumab, a PD-L1 inhibitor. They found that the pre-treatment MRI features did not accurately predict PFS and OS; however, the first post-treatment MRI features had a high predictive value for both PFS and OS. Jiang et al. [74] constructed a radiomics model to forecast the responsiveness of brain metastases from lung cancer to gamma knife radiosurgery, achieving an AUC of 0.93 in the primary dataset and 0.85 in external validation.

Despite numerous studies proving the efficacy and potential of radiomics and deep learning models in enhancing the field of neuro-oncology, they have yet to be used in clinical practice. Some of the major barriers to clinical adoption include a lack of generalizability and reproducibility between sites and scanners and lack of correlation of radiomics features with underlying biological features [75]. Recently, efforts have been made to standardize radiomics features [76,77,78,79], including by Zwanenburg et al. and the Image Biomarker Standardization Initiative [79], which accomplished the standardization of 169 radiomics features for PET, MRI, and CT. Delineating biological etiologies of radiomics features [80,81] remains an obstacle that requires further exploration. The successful application of radiomics and deep learning in clinical practice hinges on effectively addressing these challenges.

### 1.4. MR Perfusion Imaging

Blood perfusion is crucial for supplying oxygen and nutrients to tissues and is closely linked to tissue function. Therefore, disorders affecting perfusion are recognized as significant contributors to medical mortality and morbidity [82]. Evaluation of cerebral blood volume (CBV) has been extensively employed in neuro-oncological contexts, such as determining the grade of brain tumors, guiding biopsies, informing targeted therapy, and assessing disease progression and treatment response [83]. Elevated CBV is linked to heightened malignancy and proves beneficial in the grading of gliomas and prognostic assessment [84]. The connection between increased tumor aggressiveness and neovascularization has been extensively documented in the literature on brain tumor perfusion. CT and MR perfusion methods have consistently shown a correlation, indicating that higher CBVs and permeability are associated with high-grade tumors [83,85]. Previous studies found that high-grade tumors indicated statistically remarkable higher mean values than low-grade tumors [85,86]. The differentiation of low- and high-grade tumors, employing a relative cerebral blood volume (rCBV) and setting a threshold at 1.75, demonstrated a sensitivity of 95% and a specificity of 57.5% [87]. In addition, permeability values obtained through a T2-weighted technique were markedly greater for high-grade tumors compared to their counterparts in low-grade tumors [88]. The rCBV at one month could discriminate pseudoprogression arising from recurrent, progressive tumors with a specificity of 86% and sensitivity of 77% [89]. Pseudoprogression showed a lower median rCBV and permeability [90]. Similarly, the most routinely used parameter in distinguishing between tumor progression and delayed radiation necrosis is rCBV, which exhibits an elevation in recurrent tumors. In contrast, it is reduced in the vicinity of radiation necrosis [91]. 

For glioma patients without distinctive high-grade anatomical imaging characteristics, international recommendations from the European Society of Neuroradiology endorse the use of perfusion MR imaging before tissue diagnosis [92]. Additionally, MR perfusion imaging can serve in the differential diagnosis of brain tumors. Primary CNS lymphoma has shown low vascularization compared to malignant glioma, so intra-tumor CBV is not increased or only moderately increased [93]. Metastases are typically easily distinguishable from normal brain tissue, whereas glioma and lymphoma exhibit infiltrative growth patterns [94]. An elevation of CBV beyond the enhanced tumor regions indicates the infiltration zone of glioma and lymphomas, serving as evidence against metastases [95]. 

Perfusion imaging is a technique used to evaluate blood flow at the tissue level [96]. MR perfusion is performed through three primary techniques: dynamic susceptibility contrast enhancement (DSC), dynamic contrast enhancement (DCE), and arterial spin labeling (ASL). MRI contrast is administered and dynamically monitored in DCE, utilizing a T1-weighted acquisition, and in DSC, utilizing a T2*-weighted acquisition. Even though the approaches for quantifying cerebral perfusion differ, they both involve monitoring the concentration of a contrast agent over time to estimate permeability and blood volume [97]. In contrast, evaluating perfusion with ASL is accomplished without contrast, relying instead upon magnetic labeled arterial blood, while water acts as a tracer that freely diffuses.

A meta-analysis and systematic review of twenty-eight studies investigated the diagnostic presentation of both DCE and DSC in evaluating glioma after treatment. The accuracy of distinguishing treatment-induced changes from tumor recurrence is affirmed by the high sensitivity and specificity of the DSC and DCE techniques: 90% and 88% for DSC and 89% and 85 for DCE, respectively [98]. DSC perfusion is applicable for evaluating the response to treatment due to providing information on neoangiogenesis and microvascular density [99]. Permeability metrics like Ktrans (volume transfer constant), Vp (plasma volume), and Ve (extravascular extracellular space volume) obtained from DCE perfusion have been linked to microvascular leakage and vascular density. Consequently, they are employed with some success in assessing treatment response [97]. 

However, contrast agent leakage represents a pitfall to accurate analysis, and correction methods are essential for correctly evaluating CBV in brain tumors [100]. While DSC evaluation of rCBV is accurate, it may be affected by T1-weighted contrast leakage resulting from blood–brain barrier disruption. This can potentially lead to the underestimation or overestimation of rCBV values within the tumor [101]. Accordingly, some clinical trials have been performed to address this issue [102,103]. Various techniques regarding imaging acquisition, the different extracted parameters, processing software, and analysis methods have generated accurate thresholds for distinguishing tumors; e.g., a rCBV threshold range of 0.9 to 2.15 is employed in the diagnosis of tumor recurrence [98]. 

ASL represents a non-invasive approach for measuring cerebral blood flow (CBF) by utilizing labeled endogenous blood, producing a normalized CBF map as the main parameter for observation [82]. ASL has the potential to be beneficial in the extended monitoring of glioma post radiation, including those patients with renal dysfunction [104]. It was observed that the normalized CBF ratio was greater in cases of glioma recurrence in comparison with post-treatment radiation injury. Moreover, a strong linear correlation was identified between the DSC ASL and MRI approaches, with a linear regression coefficient of R = 0.85 and a significance level of *p* = 0.005. This correlation aids in differentiating recurrent glioma from radiation-related injury [105]. 

MR perfusion imaging could be an excellent diagnostic and follow-up modality in the neuro-oncology field; however, further investigations are required regarding various imaging techniques and extracted parameters.

### 1.5. Magnetic Resonance Fingerprinting

Magnetic resonance fingerprinting (MRF) has surfaced as a promising imaging method in the field of neuro-oncology, offering quantitative insights into tissue properties. MRF employs a unique single-sequence, pseudorandomized approach to generate T1 and T2 values, providing rapid quantification and tissue identification potential in neuro-oncology. It offers advantages such as accurate tumor margin delineation, distinguishing between primary and metastatic brain tumors, and discerning high-grade from low-grade gliomas. However, its efficacy in tracking longitudinal tumor progression through treatment remains unproven.

Three distinct studies were conducted to investigate MRF’s effectiveness in defining areas within solid tumors (STs), peritumoral white matter (PWM), contralateral white matter (CWM), and perilesional edema. MRF successfully distinguished solid tumor regions from CWM with T1 and T2 across three studies [106,107,108], and one study of 19 patients was able to distinguish PWM from CWM [107]. Another study found similar success in distinguishing PWM from CWM in GBM multiforme specifically [106]. However, when statistical analysis was conducted on the subset of patients with LGGs, only T1 differences were significant, with T2 trending towards significance [107]. These findings slightly differed from those made in another study, which revealed no noteworthy distinctions in either T1 or T2 values between PWM and CWM of LGGs. The same study also found that PW and CW regions did not have statistically notable variations in T1 and T2 values in metastatic brain tumors following comparison correction [106]. MRF also successfully used T1 values to separate the ST and PWM regions in LGGs. In IDH-wildtype tumors, MRF T2 and ADC values within the peritumoral edema ≤1 cm away from the ST were significantly higher than those in the ST; however, peritumoral edemas >1 cm away from ST margins were not discernable. Conversely, in mutant IDH gliomas, in the ST, MRF, T1, T2, and ADC values were markedly elevated compared to the peritumoral edema [108]. 

MRF has been tested to characterize neoplasms in three different ways: high- vs. low-grade gliomas, primary vs. metastatic brain tumors, and IDH mutant vs. wildtype gliomas. MRF displayed mixed results in distinguishing LGGs from HGGs. Two independent studies achieved successful differentiation with both T1 and T2 [107,108,109]; however, one of the two studies only showed significant T1 differences when the sample was confined to pathologically diagnosed tumors, with T2 values approaching significance [107]. Another limitation was that limited differences between solid tumor parameters were observed between GBM multiforme and LGG, except for T2 skewness, which was significant. Significant T1 and T2 variations were also observed in PWM of GBM vs. LGG. MRF has proven to be a promising tool in identifying primary vs. metastatic brain tumors. MRF mean T2 values were shown to distinguish between solid tumor of low-grade gliomas and metastases. In examining GBM multiforme versus metastases, analysis of the ST and PW regions indicated variances in T1 and T2 parameters solely prior to Bonferroni correction [106]. MRF proved effective in identifying genetic mutations, particularly differentiating IDH mutants from wildtype gliomas. Significantly higher T1 and T2 relaxation times were observed in IDH mutants for regions of interest, including solid tumor and peritumoral edema within 1 cm of solid tumor margins [108].

MRF’s non-invasive nature, devoid of radiation and wait time, makes it ideal for pediatric imaging [110]. Pediatric T1 and T2 values significantly differ across the solid tumor and peritumor regions and CWM. MRF T1 values were able to differentiate between LGG and HG while T2 values were not, paralleling MRF’s ability to characterize adult tumors [107].

Despite MRF’s diagnostic potential, limitations were noted in monitoring treatment effects. A cross-sectional study demonstrated no significant changes in T1 or T2 values between treated and untreated low-grade glioma groups. Similarly, a longitudinal assessment showed no differences before and after treatment, with a median interval of 262 days [107]. That said, CEST-MRF has recently been combined with a deep reconstruction network (DRONE) to yield much faster brain scans that are also sensitive to lower metabolite concentrations. The six-parameter DRONE reconstruction was able to produce a 256 × 256 voxel image in ~100 ms compared to the 4 h process using dictionary matching. Even under limited conditions, DRONE provided tissue maps that were less noisy compared to dictionary matching and was able to find significantly different T1 and T2 values between metastatic solid tumors and contralateral tissue [111].

Magnetic resonance fingerprinting holds great promise in neuro-oncology, offering valuable insights into tumor characterization, grading, genetic mutation identification, and pediatric imaging. Its diagnostic ability would allow physicians to quickly and non-invasively provide patients with accurate treatment plans and prognoses. However, its effectiveness in monitoring treatment responses remains inconclusive, emphasizing the need for further research. Continued exploration of MRF’s potential is essential for advancing neuro-oncological diagnostics and patient management.

### 1.6. Magnetic Resonance Spectroscopic Imaging

Magnetic resonance spectroscopy (MRS) is a type of metabolic imaging method that has the ability to detect signals generated by spins of active nuclei elements. In clinical practice, MRS signal mainly originates from hydrogen (1H) or proton-MRS, comprising water and lipid molecules, since hydrogen is one of the primary molecules in the human brain. MRS demonstrates excellent potential for evaluating brain neoplasms by supplying chemical information about different metabolites to characterize brain tumors [112]. For instance, the combination of MR spectroscopy and perfusion imaging achieved a specificity of 92% and a sensitivity of 72% in discerning between neoplasms and non-neoplastic lesions [113].

MRS is a non-invasive method for evaluating metabolic function, enabling the measurement of distinct metabolites within a specific tissue volume. In clinical evaluations using proton 1H-MRS, key measurable metabolites include N-acetyl aspartate (NAA), creatine (Cr), and choline (Cho). This established technique is widely recognized for aiding in the diagnosis and monitoring of various brain lesions [114,115]. Common metabolite changes in brain tumors include an increase in Cho, lipids, and lactate, and a decrease in Cr and NAA. Other studies also demonstrated the use of the MRS technique as a powerful method in discerning metabolic changes linked to tumor grading and progression. Especially, a depression of NAA with an elevation in Cho are suggested as a reliable marker of tumor characterization [87,116]. In neuro-oncology, achieving complete tumor excision is the main therapeutic objective. Hence, it is essential to accurately identify the precise boundaries of the tumor. Proton MRS guides the surgeon in the evaluation of regions with high metabolic activity (low NAA levels and elevated Cho levels) for biopsy [117,118].

The increase in Cho is due to proliferation and cell membrane turnover [119,120]. Cho levels differ markedly based on cellular density, tumor grade, and necrosis. Cho resonance is particularly prominent in areas characterized by elevated neoplastic density which is noticeably lower in moderate- to low-grade tumors [121,122]. NAA serves as a neuronal indicator, and its concentration diminishes as a result of neuronal damage, as observed in conditions such as extensive lesions, hypoxia, dementia, or multiple sclerosis. The connection between the decrease in NAA concentration and the increase in glioma grading related to a reduction in neuronal density makes NAA a possible substantial diagnostics marker for glioma [123]. Several studies have shown that the MRS technique can potentially evaluate cerebral glioma grading accurately [124,125,126]. The most prevalent primary tumor in the central nervous system originating from glial cells is glioma. In classical histological analyses, gliomas can be categorized into high-grade and low-grade through atypia, anaplasia, mitosis, necrosis, and microvascular proliferation. The role of 1H-MRS imaging to forecast the survival rate of GBM patients has been evaluated in brain tumor populations [127,128]. 

Histopathological findings with data assessed using MRS in patients with recurrent or new glioma clarified that decreased NAA and increased Cho were more associated with tumors than normal brain parenchyma and necrosis [129]. Quantitative or qualitative detection of elevated Cho/NAA peak height ratios serves as a predictive factor in diagnosing high-grade glioma [130,131]. Furthermore, lipid/lactate in untreated glioma indicates the diagnosis of necrotic grade IV tumor [131,132]. In another study, the ability to differentiate biopsy samples containing glial tumors from non-tumoral regions containing a combination of normal, gliotic, edematous, and necrotic tissue exhibited a sensitivity of 90% and specificity of 86% when employing a Cho–NAA index (CNI) threshold of 2.5 [133]. Proton MRS has been utilized to differentiate between tumor recurrence and radiation-induced tissue damage following radiation and gamma knife radiosurgery. Elevated Cho signal, Cho/Cr, or Cho/NAA ratios are indicative of recurrence, whereas diminished Cho and Cr levels suggest radiation-induced necrosis [134]. Post radiotherapy or gamma knife radiosurgery, a decrease in Cho levels may signify partial remission, whereas stability or an increase in Cho suggests disease progression [135]. Combining short and long TE MRS gives a diagnostic validity of 98% for the main pediatric brain tumor types, such as medulloblastoma, ependymoma, and pilocytic astrocytoma [136]. Moreover, the percentage alteration in the Cho/NAA ratio detected through proton MR spectroscopic imaging proved beneficial in prediction of tumor advancement in pediatric brain tumor cases [137]. Furthermore, an elevated Cho/NAA ratio was linked to reduced survival rates in children experiencing recurrent glioma [138]. 

Myo-inositol (MI) is a cellular osmotic regulator which is detectable within the brain via short TE MRS. There are fluctuations in its concentration within brain tumors. High-grade tumors such as GBM have lower levels resulting from disruptions in the blood–brain barrier and may lead to disturbances of the osmotic equilibrium [139,140]. MI normalized by contralateral creatine (MI/c-Cr) values could serve as an indicator aiding in the prediction of responses to anti-angiogenic treatment and differentiation between individuals with short-term and long-term survival [141,142]. Reduced MI/c-Cr levels in intra-tumoral, contralateral, and peritumoral volumes may indicate a prognosis of poor survival and lack of response to anti-angiogenic therapy before initiating treatment of recurrent GBM [141,143]. A recent study proved that MI/c-Cr has the capability to differentiate between pseudo- and true progression, highlighting the significance of this MRS metabolite with a short echo time [144]. 

In the 2021 WHO tumor classification, the existence of the isocitrate dehydrogenase (IDH 1/2) enzyme mutation is what distinguishes astrocytoma from GBM, further highlighting the clinical role of 2-hydroxyglutarate (2-HG) MRS [145]. IDH mutations, predominantly observed in oligodendroglia and astrocytic tumors, have been identified as a marker for low-grade glioma. Gliomas with IDH mutations exhibit improved treatment responses and longer survival durations compared to tumors with IDH-wildtype [146]. Mutations in the IDH 1/2 enzyme, commonly found in grade II and grade III gliomas, cause the accumulation of 2-HG in brain tumor cells [147]. Accordingly, 2-HG can be a valuable biomarker and onco-metabolite for diagnosing and observing therapy responses in IDH-mutated gliomas. MRS can detect this metabolite at a high field strength [148,149,150]. In one study, 1H-MRS with a short echo time accurately identified the presence of an IDH mutation with an accuracy of 88.39%, sensitivity of 76.92%, and specificity of 94.52% [151].

Utilizing MRS imaging along with conventional MRI can reveal essential information concerning the biological traits of tumors to assist effective treatment of recurrent GBM [141]. In patients undergoing chemotherapy, proton MRS could offer insights into the functional response regarding tumor chemosensitivity and early treatment modification to prevent unnecessary toxicity [152]. Non-invasive accurate diagnosis of glioma and recurrent glioma is vital, as the prognosis and therapeutic plans mainly rely on the histopathological grade of the tumor. Proton MRS imaging along with other combined imaging approaches can provide valuable data and assist the surgeon in acquiring representative cancer samples for histological examination and resection by pinpointing active tumor regions. As elucidated earlier, MRS offers valuable potential information for targeted radiotherapy and selecting the optimal patient treatment. 

### 1.7. Magnetic Resonance Elastography

Magnetic resonance elastography (MRE) is a non-invasive method for measuring the mechanical characteristics of tissues. Brain tumor cells and their extracellular matrix demonstrate altered tissue mechanics which manifests in varied tissue stiffness. A prior knowledge of the visco-elastic property of brain tumors may guide neurosurgeons in the pre-operative planning of optimal surgical techniques and therapeutic stratification of patients. Studies in GBM have also shown that MRE may provide information on the WHO grade and IDH status of the tumor, where higher-grade gliomas and IDG wildtype tumors were softer than lower-grade and IDH mutant tumors. MRE may therefore significantly contribute to the growing field of “mechanogenomics” [153].

### 1.8. Intra-Operative Ultrasound

While other modalities that are used to characterize neuro-oncological pathology are mostly pre-operative in nature, the intra-operative setting provides its own set of challenges. For example, as the brain is a non-fixed structure, “brain shift” often occurs and can be due to a variety of factors including surgical hardware manipulation, gravity, and fluid loss [154]. This can cause incongruity between pre-operative imaging and actual surgical visualization, making accurate surgical margins difficult to appreciate. Ultrasound, as a real-time intra-operative imaging modality, has been utilized to address these issues. With the improvement in probe technology and development of advanced software, intra-operative ultrasound (ioUS) in neuro-oncological surgeries is increasingly being utilized in the operating room. Most literature on intra-operative ultrasound has emerged in the past decade, particularly in the past few years, including a few review articles by Dixon et al. [155] and Moiyadi [156], a clinical trial by Incekara et al. [157], and a textbook written by Prada et al. [158].

The primary benefit of ioUS lies in its capability to offer imaging in real time during surgical resection in defining surgical borders when compared to pre-operative imaging, which is often fused with ultrasound imaging [159]. While MRI can also be utilized intra-operatively, only ioUS provides real-time imaging. Intra-operative MRI (iMRI) has significant disadvantages, such as cost and increased operative time, that ioUS generally does not have [160,161,162,163]. Oftentimes, iMRI and ioUS are used in conjunction through fusion imaging, and have been shown to correct for brain shift in a study involving 58 patients with 42 cases successfully correcting for the commonly encountered issue [162]. On the other hand, ioUS used alone can provide similar results to iMRI. Studies involving pediatric patients reported a high concordance between ioUS and post-operative MRI and an equal efficacy of iMRI and ioUS in regards to determining the extent of brain tumor resection [164]. 

While ioUS has its advantages, its constraints still limit its widespread adoption in the intra-operative setting [155]. Ultrasound artifacts, such as acoustic shadowing and posterior wall acoustic enhancement, limit evaluation. In addition, the field of view is confined to the craniotomy site as ultrasound cannot penetrate the nearby intact calvarium. Another limitation is that ultrasound remains operator-dependent, with variations in technique, and is significantly more difficult to standardize in imaging, interpretation, and teaching [165]. 

Specific advanced ultrasound modalities, e.g., contrast-enhanced ultrasound and elastography, have been described for neuro-oncologic purposes. In contrast-enhanced ultrasound, microbubbles allow for the visualization of surrounding arteries and veins, characterize tumor microvascularization, and allow for better definition of tumor borders, especially in tumors with ill-defined boundaries on B-mode [166]. Similar to liver elastostography in the evaluation of liver stiffness in cirrhosis, ultrasound can be used intra-operatively to determine certain tumor characteristics based on stiffness and detect residual tumor tissue [158,167]. 

While ioUS is spatially inferior to CT and MRI, with lower resolution, it has its own set of advantages that make it a valuable tool in the operating room. This intra-operative modality allows for the real-time visualization of tumor margins, surrounding structures, and nearby vasculature, allowing for safer resections and more accurate planning. In addition, there are significant cost- and time-saving benefits. However, utilization is still relatively new, and its main limitations are the lack of standardization in training and imaging techniques and the dependency on the user. Generally, high-grade gliomas have been found to be more echogenic than low-grade gliomas; however, the sonographic appearance of different brain tumors is highly variable due to a variety of factors requiring correlation with pre-operative imaging [155]. Newer literature has attempted to standardize ioUS [155]. For example, studies have attempted to identify the pre-operative parameters that would indicate the need for ioUS [168]. Still, with improvements in ultrasound technology and increasing utilization, ioUS holds substantial promise as a tool that will be increasingly implemented in the future.

## 2. Conclusions

In conclusion, the field of neuro-oncological imaging has made significant progress in recent years, revolutionizing our approach to the diagnosis, staging, management, and monitoring of brain and CNS tumors. With the advent of cutting-edge imaging modalities and techniques, we are incrementally achieving a deeper understanding of the complicated nature of these diseases and their response to treatment.

These advancements have not only improved the accuracy of tumor diagnosis but have also addressed challenging clinical scenarios, including the evaluation of treatment-related changes, responses to novel therapies like immunotherapy, and the early detection of disease progression. The knowledge of both the capabilities and limitations of these emerging imaging technologies is essential for providing a higher level of personalized care to patients with neuro-oncological conditions.

Overall, the future of neuro-oncological imaging is promising, and continued investigation in this field will lead to further advances in improving outcomes for patients with CNS tumors.

## Figures and Tables

**Figure 1 cancers-16-00576-f001:**
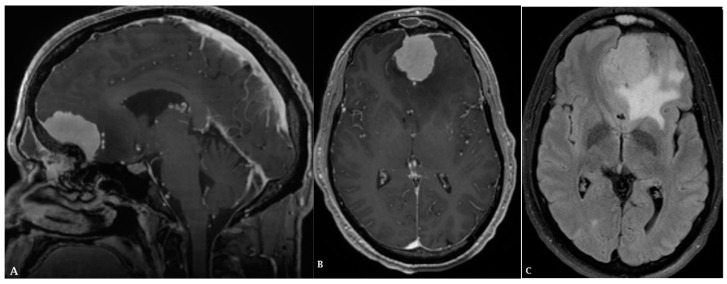
A forty-eight-year-old male presented with a solid homogeneously enhancing left frontal lesion with dural tail sign and significant perilesional vasogenic edema, in keeping with WHO grade I meningioma ((**A**,**B**): T1 post contrast, (**C**): FLAIR).

**Figure 2 cancers-16-00576-f002:**
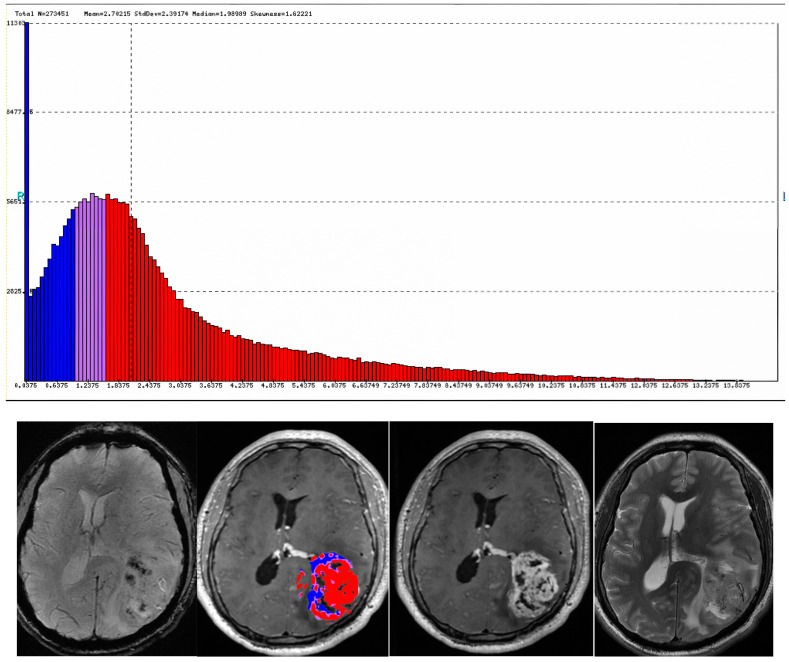
A thirty-year-old male was found to have a heterogeneously enhancing left parietal mass with perilesional vasogenic edema, resulting in significant compressive effect on the left lateral ventricle and shift of midline to the right. The lesion was diagnosed as wildtype GBM WHO grade IV.

**Figure 3 cancers-16-00576-f003:**
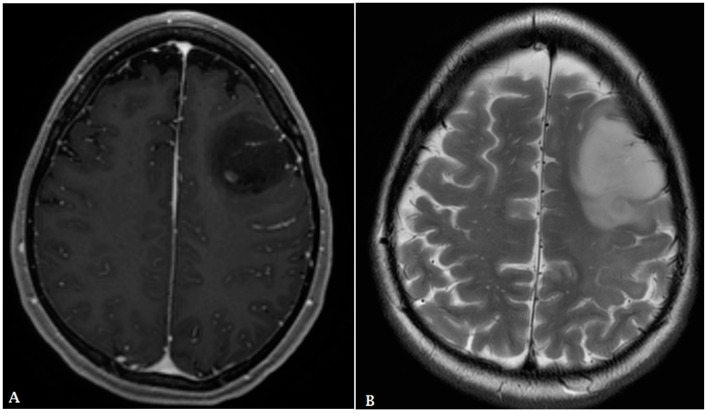
A twenty-nine-year-old female presented with a heterogeneously enhancing subcortical lesion involving the left frontal lobe with perilesional vasogenic edema. Arterial spin-labeling (ASL) was negative. The lesion was found to be a grade III IDH1 mutant astrocytoma on pathology. ((**A**): post-contrast T1, (**B**): T2, (**C**): FLAIR, (**D**): ASL).

**Figure 4 cancers-16-00576-f004:**
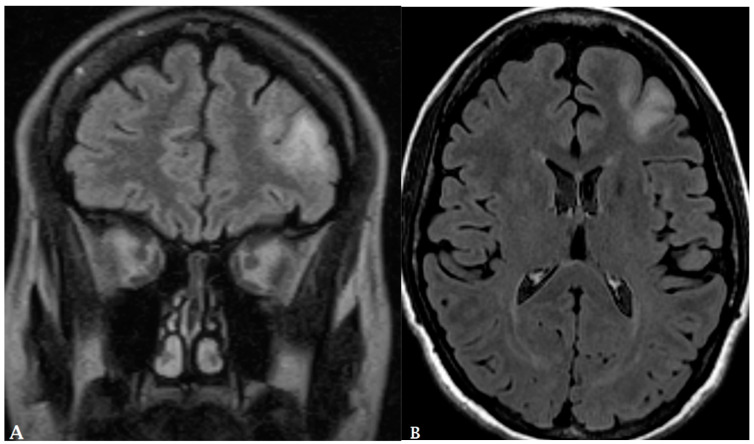
A 30-year-old female presented with subcortical ill-defined vasogenic edema (**A**,**B**). Pathologic evaluation demonstrated grade II IDH1 mutation ATRX wildtype left frontal astrocytoma.

**Figure 5 cancers-16-00576-f005:**
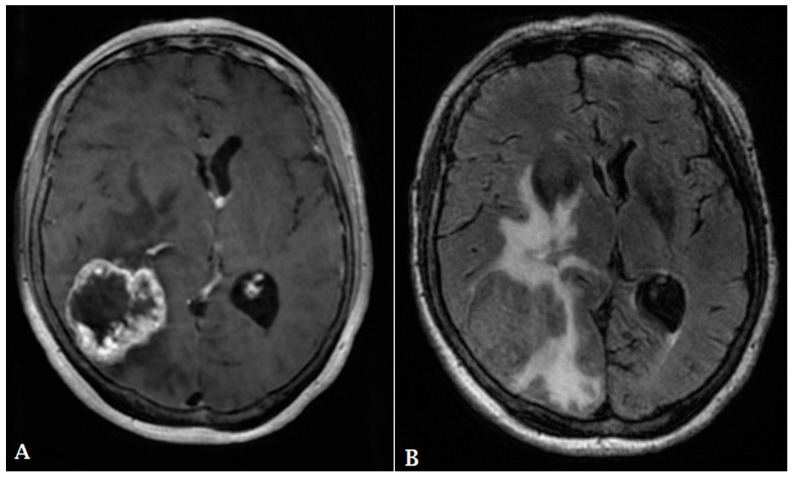
A 60-year-old male with history of colon cancer presented with heterogeneously enhancing mass involving the right parietal lobe ((**A**): T1 post contrast, (**B**): FLAIR). There is significant periolesional vasogenic edema and mass effect on the right lateral ventricle, resulting in midline shift to the left. The findings are consistent with cerebral metastasis.

**Figure 6 cancers-16-00576-f006:**
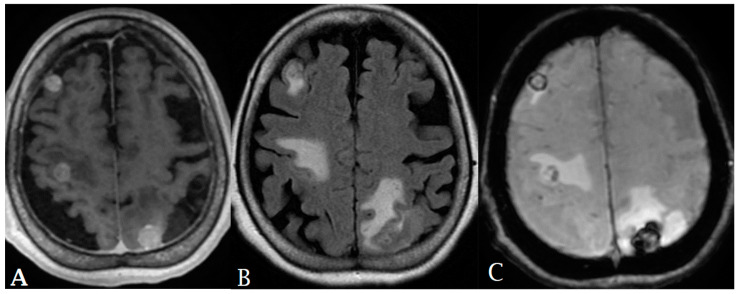
A 65-year-old female with history of papillary thyroid cancer was found to have multiple small enhancing cerebral metastases (**A**) with perilesional vasogenic edema (FALIR (**B**)) and hemorrhagic component (SWI (**C**)).

**Figure 7 cancers-16-00576-f007:**
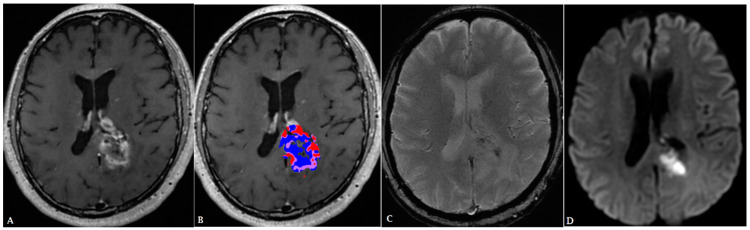
A 55-year-old male with history of renal cell carcinoma presented with heterogenoeusly enhancing mass (T1 post contrast (**A**)). Post-treatment perfusion map (**B**) demonstrates two components: blue areas are suggestive of post-treatment changes/radiation necrosis and red areas are indicative of minimal residual tumor. The lesion demonstrates internal diffusion restriction (**C**,**D**).

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
