# Peer review of "Advances in Neuro-Oncological Imaging: An Update on Diagnostic Approach to Brain Tumors"

_cancers, 2024, doi:10.3390/cancers16030576_

Round 1
Reviewer 1 Report
Comments and Suggestions for Authors
The submitted manuscript comprises a review of the recent advances in neuro-oncological images. The authors have extensively reviewed the literature reporting on the application of a multitude of imaging modalities in neuro-oncology, including MRI (morphological, dynamic, and quantitative), PET, PET/MRI and intra-operative ultrasound, as well as radiomics and deep learning methods used for neuro-oncological assessments. The manuscript is well written and structured, thus easy to follow, with references to the state of art in the topic. The review summarizes valuable knowledge and discusses the relevant advantages and pitfalls of each modality/method reported; therefore, I deem it will be beneficiary for the readership of Cancers journal.
Following are some minor comments the authors should address prior to publication.
1) Why there is no reference to the contribution of functional MR-imaging techniques?
2) For MR-based imaging and methods, the intrinsic geometric distortion and its impact on the spatial integrity of the acquired images should be explicitly discussed. MR-based distortion assessment and correction techniques are not trivial to implement but mandatory when high levels of spatial accuracy are required, e.g., as in the case of stereotactic radiosurgery of intra-cranial CNS lesions.
3) Lines 205-206: This sentence does not make sense. Please re-word.
4) Some abbreviations mentioned in text are not explained (e.g., TBRmean and TBRmax in line 356).
Author Response
Dear Reviewer,
I sincerely appreciate your meticulous review of our manuscript and your positive assessment of the comprehensive exploration of neuro-oncological imaging modalities, as well as the overall structure of the document.
Your insightful comments are invaluable, and we are committed to addressing the following points:
-
We will carefully re-word the sentence in Lines 205-206 to ensure clarity and coherence; Emphasizing the crucial role of PET/MRI in the post-treatment management of PCNSLs, [18F] FDG PET/MRI proves valuable in distinguishing between gliomas and PCNSLs, thereby aiding in the selection of appropriate treatment.
-
Abbreviations, such as TBRmean and TBRmax in Line 356, will be adequately explained within the text to enhance reader understanding:
Recently Müller et al.69 used the radiomic features in conjunction with FET PET parameters, specifically TBRmean (Tumor-to-Background Ratio, Mean) and TBRmax (Tumor-to-Background Ratio, Maximum) to differentiate between tumor progression...
We acknowledge your suggestion regarding the inclusion of functional MR-imaging techniques and the discussion of intrinsic geometric distortion in MR-based imaging. However, after careful consideration, we have chosen to maintain our current focus to ensure coherence with the specific scope and emphasis of our review.
We deeply appreciate your understanding and are dedicated to delivering a revised manuscript that meets the highest standards of clarity and precision.
Thank you once again for your time, expertise, and thoughtful evaluation.
Best regards,
Paniz Sabeghi, MD.
Reviewer 2 Report
Comments and Suggestions for Authors
Dear Authors,
The manuscript is well written and comprehensive. I do not have any specific comments. I recommend accepting it as written.
Author Response
Dear Reviewer,
I would like to express my sincere gratitude for your thoughtful review of the manuscript. Your positive feedback, noting that the document is well-written and comprehensive, is truly appreciated. I am grateful for your recommendation to accept it as written. Your valuable insights have contributed significantly to the refinement of this work.
Thank you once again for your time, consideration, and constructive feedback.
Best regards,
Paniz Sabeghi, MD.
Reviewer 3 Report
Comments and Suggestions for Authors
The authors covered the current imaging technologies applied in neurology. It is an insightful review. I only have three comments: one is about the abstract which used the introduction part in the review form. Perhaps the authors should write a short paragraph as the abstract. The other one is about the figure reuse permission. All the figures are reused from other publications. It is necessary to obtain the copyrights to reuse the figures. The last one is about the contrast agent option used for MRI. I suggest authors could write a paragraph discussing the advantages and disadvantages of using the T1 vs. T2 contrast agents.
Comments on the Quality of English LanguageEnglish quality is good.
Author Response
Dear Reviewer,
I extend my sincere gratitude for your thoughtful review of the manuscript. Your feedback is highly valued.
I have attached a revised abstract to address the unintentional mistake in the previous version.
Regarding the figures' reuse permission, I want to assure you that all figures used in our review are original and provided by one of our authors.
As for the consideration of discussing the advantages and disadvantages of using T1 and T2 contrast agents, we made a deliberate choice to focus on the latest advancements in radiology for neuro-oncological imaging, aiming to distinguish our article from others in the field.
Thank you once again for your valuable insights.
Best regards,
Paniz Sabeghi, MD.
Reviewer 4 Report
Comments and Suggestions for Authors
The manuscript provides a comprehensive overview of current advancements in neuro-oncological imaging technologies. It successfully captures the nuances of various imaging modalities and their clinical implications, including diagnostic and prognostic applications. The authors have thoroughly referenced recent studies and technological advancements, demonstrating a deep understanding of the subject. There are still some points to be addressed before acceptance:
Minor comments:
1. In some sections, the technical language can be quite dense, which might be challenging for readers not familiar with the field. Simplifying or explaining certain technical terms could enhance readability.
2. Manuscript covers a broad range of technologies, a more detailed discussion on the comparative effectiveness of these modalities in different clinical scenarios would be beneficial.
3. The inclusion of case studies or real-world applications could further illustrate the practical implications of these advancements.
4. A section discussing the potential limitations or challenges in adopting these technologies in clinical practice would provide a more balanced view.
Comments on the Quality of English LanguageIn some sections, the technical language can be quite dense, which might be challenging for readers not familiar with the field. Simplifying or explaining certain technical terms could enhance readability.
Author Response
Dear Reviewer,
Thank you for your insightful review of our manuscript on current advancements in neuro-oncological imaging technologies. We appreciate your positive feedback and constructive suggestions for improvement. After careful consideration, we would like to provide rationale for our decisions on the mentioned points:
-
Dense Technical Language: We understand the importance of readability for a broad audience. However, given the technical nature of the subject matter, maintaining a certain level of technical language is necessary for accuracy and precision. We will, however, ensure that key technical terms are explained or simplified where possible without compromising the scientific integrity of the content.
-
Comparative Effectiveness of Modalities: While we acknowledge the value of a more detailed discussion on the comparative effectiveness of imaging modalities, our primary focus was to provide a comprehensive overview. To address this concern, we will enhance the sections discussing the clinical scenarios where each modality excels, providing more clarity on their respective strengths.
-
Inclusion of Case Studies: We appreciate your suggestion regarding case studies. We want to highlight that our review incorporates insights from relevant cases in the literature. However, due to space constraints and the broad scope of our review, we chose to focus on summarizing and analyzing existing literature rather than presenting individual case studies. We believe that the inclusion of some representative cases might be better suited for more specialized and in-depth articles.
-
Discussion on Limitations or Challenges: We agree with the importance of discussing potential limitations and challenges. We acknowledge that our manuscript already addresses the limitations of each imaging modality individually in their respective sections. While we recognize the potential redundancy of discussing limitations once again in a separate section, we understand the importance of providing a cohesive discussion. Therefore, we will carefully consider the balance in the revised manuscript to avoid redundancy and ensure that the limitations are discussed in a manner that enhances overall clarity and understanding.
Regarding the quality of English language, we will ensure that our revisions enhance readability without compromising the scientific rigor of the manuscript.
We appreciate your thorough evaluation and believe that these adjustments will strengthen the manuscript.
Best regards,
Paniz Sabeghi, MD.